analytical chemistry

Ag$_2$S quantum dots, carbon dots, ratiometric fluorescent probe, pH Detection

**Authors for correspondence:**
Yuan Wu
e-mail: yuanwu@mail.hzau.edu.cn
Lu Chen
e-mail: chenlu@mail.hzau.edu.cn

[†]These authors contributed equally to this work.

This article has been edited by the Royal Society of Chemistry, including the commissioning, peer review process and editorial aspects up to the point of acceptance.

# A ratiometric fluorescent probe for pH detection based on Ag$_2$S quantum dots– carbon dots nanohybrids

Xiaoxue Lei[†], Yiying Fu[†], Yuan Wu, Lu Chen and Jiangong Liang

College of Science, Huazhong Agricultural University, Wuhan 430070, People's Republic of China

YW, 0000-0002-6953-791X

In this study, a novel ratiometric fluorescent nanoprobe for pH monitoring has been developed by synthesizing red fluorescent Ag$_2$S quantum dots (Ag$_2$S QDs) and green fluorescent carbon dots (CDs) nanohybrids (Ag$_2$S CDs) in one pot using CDs as templates. The nanoprobe exhibits dual-emission peaks at 500 and 670 nm under a single-excitation wavelength of 450 nm. The red fluorescence can be selectively quenched by increasing pH, while the green fluorescence is an internal reference. Therefore, the change of the relative fluorescence intensity ($I_{500}/I_{670}$) in the ratiometric Ag$_2$S CDs probes can be used for pH sensing. The results revealed that $I_{500}/I_{670}$ of Ag$_2$S CDs probes was linearly related to pH variation between pH 5.4 and 6.8. Meanwhile, the Ag$_2$S CDs probes possessed a good reversibility along with pH changing between 5.0 and 7.0 without any interruption from common metal ions, proteins and other interferences.

## 1. Introduction

Quantum dots (QDs) have attracted wide attention in various research areas due to their unique properties, such as highly photochemical stability, broad excitation and size-tunable optical spectra, high extinction coefficients, high quantum yields and narrow emission peaks [1]. High-quality Ag$_2$S QDs have been extensively developed as chemo/biosensors [2–5]. Generally, these Ag$_2$S QD-based sensors are based on the quenching or enhancement of single-fluorescence intensity signal, which are not suitable for real samples, because the fluorescence intensity of the single emission sensors fluctuates along with instrumental efficiency and environmental conditions [6].

To solve this defect, ratiometric fluorescent sensors using two individual fluorescence emission wavelengths were developed to

**Scheme 1.** The working principle for the dual-emission fluorescence nanohybrids towards pH probe.

achieve self-calibration [7,8], which is because one emission peak maintains constant as internal reference and the other is sensitive to the analyte as signal report unit. Simultaneously, the instrumental error can be effectively eliminated, and the sensitivity of the sensor can be enhanced [9,10].

Continuous online monitoring of pH is considerably significant for bioprocess and environment [11]. Until now, small molecule probes have been developed to monitor pH using ratiometric fluorescence changing with broad pH response over a range of 1.0 pH unit [12,13]. Several QD-based pH sensors using ratiometric fluorescent sensing have been reported, in which organic dyes were chemically conjugated with QDs as internal reference [14,15]. However, organic dyes used in these projects commonly suffer from fast-photobleaching. Thus, choosing one stable emitter as internal reference is of great importance for developing ratiometric fluorescent sensor. Carbon dots (CDs) are a fascinating nano-fluorescent material with superior properties including stable photostability, simple synthesis and modification, great biodegradability and biocompatibility [16–18], which make them perfect candidates as internal reference in comparison with organic dyes for developing ratiometric fluorescent sensor. Moreover, the photoluminescence of CDs can be tuned between UV to near infrared regions of the spectrum according to the particle size, synthesis precursors and the surface functional group [19,20]. Naccache and co-workers [21] developed ratiometric detection of heavy ions using dual-emitting CDs.

Herein, we have prepared the $Ag_2S$ CDs nanohybrids, which combine the fluorescence property of $Ag_2S$ QDs and CDs, to develop a novel ratiometric fluorescent sensor for pH sensing. These $Ag_2S$ CDs nanohybrids possess two emission peaks at 500 and 670 nm under a single excitation of 450 nm. The red fluorescence of the $Ag_2S$ CDs can be quenched by increasing pH, while the green fluorescence of the $Ag_2S$ CDs is insensitive to pH. The variation of fluorescence intensity ratio ($I_{500}/I_{670}$) is linearly related to pH between 5.0 and 7.0, indicating the successful development of pH-responsive ratiometric fluorescent probe. The disadvantage of this probe is narrow pH response. However, it shows fluorescence change within a 0.2 pH unit change between 5.0 and 7.0 with satisfied linear relationship. It is known even with pH changes below 1.0 unit, our environments and physiology will be significantly impacted. Hence, our probes which sharply respond to small pH changes within a narrow range are desired.

## 2. Results and discussion

The working principle for the dual-emission fluorescence nanohybrids towards pH probe is illustrated as scheme 1. $Ag_2S$ CDs nanohybrids showing dual-emission peaks at 500 and 670 nm under a single-excitation wavelength of 450 nm were prepared by using CDs as templates to develop the ratiometric fluorescent probes, in which green emission can be adopted as the reference signal due to their excellent photostability towards pH. The red emission of $Ag_2S$ CDs acts as the signal report unit for pH, because the fluorescence at 670 nm is effectively quenched by increasing pH. Therefore, the change of relative fluorescence intensity ($I_{500}/I_{670}$) in the ratiometric $Ag_2S$ CDs probes is suitable for detection of pH.

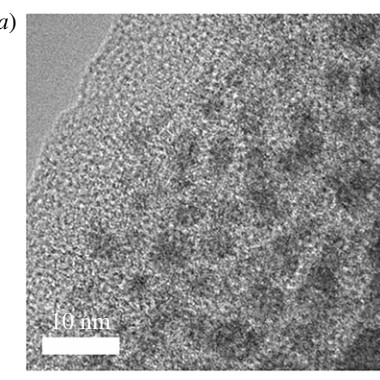 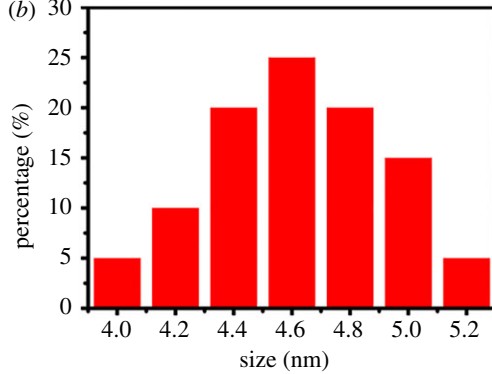

**Figure 1.** (*a*) HR-TEM and (*b*) DLS of Ag₂S CDs.

The morphology and size of Ag$_2$S QDs and Ag$_2$S CDs were characterized by high-resolution transmission electron microscope (HR-TEM) and dynamic light scattering (DLS). As shown in figure 1$a$,$b$, the as-prepared Ag$_2$S CDs were well dispersed with a uniform size of $4.7 \pm 0.4$ nm, which was much larger than that of Ag$_2$S QDs with a clear spherical shape ($1.8 \pm 0.4$ nm) (electronic supplementary material, figure S1). The zeta potential results of CDs, Ag$_2$S QDs and Ag$_2$S CDs are as shown in electronic supplementary material, figure S2. The zeta potential of CDs was found to be $+3.71 \pm 0.53$ mV, illustrating the positive surface charge of carbon dots, and the zeta potential of Ag$_2$S QDs was discovered to be $-50.90 \pm 0.90$ mV, manifesting the negative surface charge of Ag$_2$S QDs. The as-prepared Ag$_2$S CDs consisting of Ag$_2$S QDs and CDs was $-42.40 \pm 1.10$ mV with negative charge. The zeta potential value of Ag$_2$S CDs was lower than that of the Ag$_2$S QDs.

The surface functional groups of Ag$_2$S CDs were identified by Fourier transform infrared spectroscopy (FTIR). As shown in electronic supplementary material, figure S3, the characteristic absorption bands centred at 3426, 1635 and 1230 cm$^{-1}$ are ascribed to the stretching vibrations of O-H, C=O and C-O, respectively, indicating the existence of -COOH group [22,23]. The peak at 1527 cm$^{-1}$ is assigned to N-H bending vibrations. The bands at 1400 and 1311 cm$^{-1}$ can be attributed to amide II and amide III, suggesting the presence of $-$NH$_2$ group [24]. These data demonstrated that the Ag$_2$S CDs were encompassed with $-$COOH and $-$NH$_2$ groups. The elemental quantification and chemical bond analysis of Ag$_2$S CDs were performed by X-ray photoelectron spectroscopy (XPS). The results are shown in figure 2. The peaks at 161.7 and 162.7 eV are the characteristic binding energy of S 2p, which are attributed to Ag-S and S, respectively (figure 2$a$) [25]. The peaks at 367.8 and 378.3 eV are in accordance with the binding energy of Ag 3d$_{5/2}$ and Ag 3d$_{3/2}$ electrons of metallic Ag(0) (figure 2$b$) [26]. The S 2p and Ag 3d spectra of XPS indicate the formation of Ag$_2$S QDs. The high-resolution XPS spectrum of C 1s can be interpreted as four different components at 284.7, 285.9, 287.6 and 288.9 eV, corresponding to -CH$_2$-CH$_3$, -CH-, -CONH$_2$, and -COOH groups, respectively (figure 2$c$) [27]. The N 1s peaks at 399.3 and 400.9 eV, confirmed the existence of $-$NH and $-$NH$_3^+$, respectively (figure 2$d$) [28,29]. These XPS results were consistent with that of FTIR spectrum, further demonstrating the presence of $-$COOH and $-$NH$_2$ groups on the surface of Ag$_2$S CDs.

Ag$_2$S CDs emitted bright pink fluorescence under a UV lamp while the fluorescence turned to yellow under visible light irradiation (figure 3$a$). In order to excite the CDs and Ag$_2$S QDs in the as-prepared Ag$_2$S CDs simultaneously, we found that the best excitation wavelength for Ag$_2$S CDs is 450 nm. It was serendipitously found that the fluorescence of Ag$_2$S CDs obviously changed from pink to green under the UV lamp when the environmental pH increased from 5.0 to 7.0 (figure 3$b$). Furthermore, the UV absorption intensity of Ag$_2$S CDs gradually increased with decreasing pH (figure 4$a$), which both inspired us to develop pH sensor based on the as-prepared Ag$_2$S CDs. The normalized fluorescence spectra of the CDs, Ag$_2$S QDs and Ag$_2$S CDs are shown in figure 4$b$. The CDs exhibited a maximum emission peak at 502 nm, while that the Ag$_2$S QDs were at 670 nm. The Ag$_2$S CDs show dual-emission peaks at 500 and 670 nm under a single excitation of 450 nm, which provided the theoretical basis for developing ratiometric pH probes based on the as-prepared Ag$_2$S CDs.

To achieve the ideal fluorescence property of Ag$_2$S CDs, the concentration of CDs used in synthesizing Ag$_2$S CDs was optimized. The result is as shown in figure 5$a$, and 25.5 µg ml$^{-1}$ was chosen as the optimum CDs concentration for the synthesis of Ag$_2$S CDs. Because the emission peak at 500 nm from CDs was used as the reference signal, the emission peak at 670 nm was used as the signal report unit for pH. Hence, the fluorescence intensity at 500 nm should not be stronger than that at 670 nm. The response of the

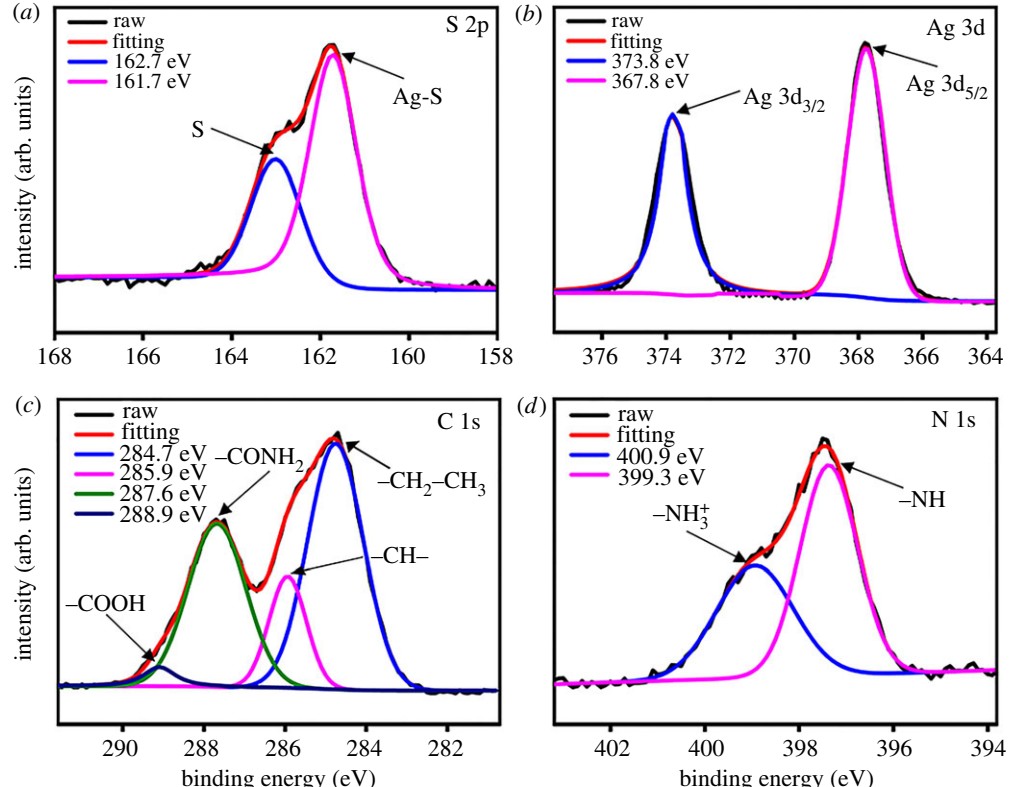

**Figure 2.** The high-resolution XPS spectra of Ag$_2$S CDs. (*a*) S 2p, (*b*) Ag 3d, (*c*) C 1s and (*d*) N 1s.

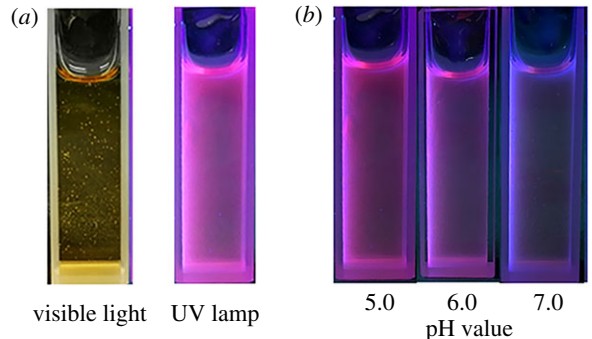

**Figure 3.** (*a*) The images of Ag$_2$S CDs under irradiation of visible light (left) and UV light (right, 365 nm); (*b*) The images of Ag$_2$S CDs with pH value from 5.0 to 7.0 (from left to right) under the UV light.

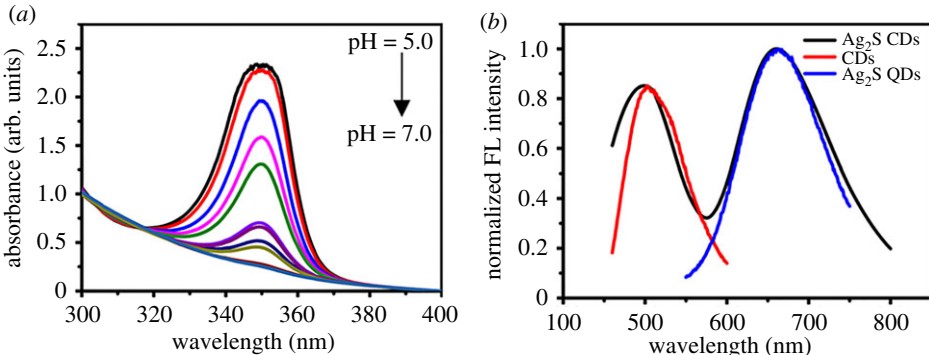

**Figure 4.** (*a*) The UV-vis absorption spectra of Ag$_2$S CDs within different pH values; (*b*) The normalized fluorescence (FL) emission spectra of CDs (red), Ag$_2$S QDs (blue) and Ag$_2$S CDs (black). The concentration of the nanoparticles in each test is 0.02 mg ml$^{-1}$.

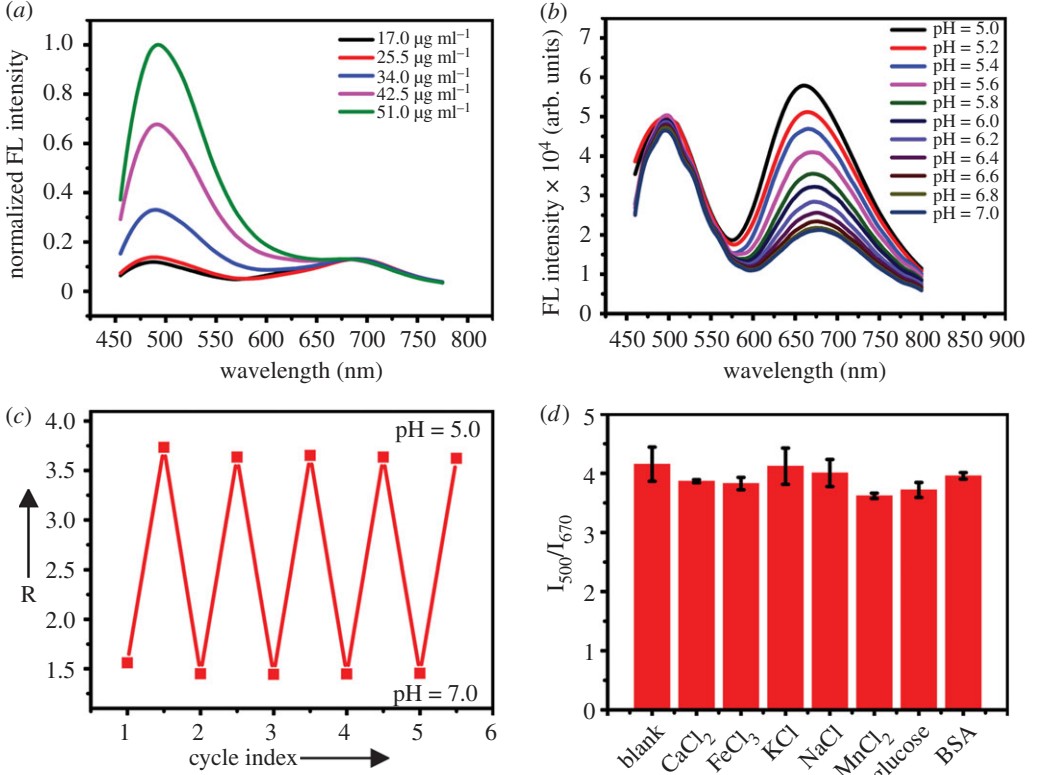

**Figure 5.** (a) The fluorescence spectra of Ag$_2$S CDs under different concentrations of CDs; (b) The fluorescence emission spectra of the Ag$_2$S CDs under different pH values; (c) pH reversibility study of Ag$_2$S CDs between pH 5.0 and 7.0; (d) The effects of different substances on Ag$_2$S CDs. The concentration of the Ag$_2$S CDs is 0.02 mg ml$^{-1}$.

**Table 1.** Comparison of this method with some previously reported method for pH sensing in ratiometric method.

| method | $\lambda_{ex}$/nm | $\lambda_{em}$/nm | range[a] | ref[b] |
|---|---|---|---|---|
| small molecule probe | 405 | 588/531 | 6.35–8.0 | [9] |
| fluorescent nanogel loaded with fluorophores | 450 | 500/620 | 4.9–9.0 | [33] |
| water-soluble cationic porphyrin derivatives | 471 | 650/719 | 2.1–8.0 | [34] |
| mesoporous silica nanoparticles with rhodamine dye | 488 | 514/550 | 3.5–7.5 | [35] |
| Ag$_2$S CDs nanohybrids | 450 | 500/610 | 5.0–7.0 | this work |

[a]Linearity range of pH detection.
[b]Reference.

dual-emission Ag$_2$S CDs probes towards pH was performed to prove the successful development of the ratiometric fluorescent probes, as demonstrated in figure 5b. With a single excitation of 450 nm, the fluorescence intensity of Ag$_2$S CDs at 670 nm was gradually weakened with increasing of pH, while the fluorescence intensity of Ag$_2$S CDs at 500 nm remained constant as the internal reference, which is available for the ratiometric detection of pH. As shown in electronic supplementary material, figure S4, the ratio of the fluorescence intensity (I$_{500}$/I$_{670}$) gradually decreased with increasing pH and the I$_{500}$/I$_{670}$ ratio showed linearity with different pH values in the range of 5.4–6.8, and the linear regression equation was R = −3.40 + 0.83pH with a linear correlation coefficient of 0.997. Ag$_2$S QDs and CDs physically mixing cannot result in the change of the fluorescence intensity only at 670 nm with increasing of pH, while the fluorescence intensity at 500 nm remained constant. Hence, it could be confirmed that the ratiometric pH probes were based on hybrid structure of Ag$_2$S CDs. The mechanism for the pH-responsive Ag$_2$S CDs probes was considered to the surface protonation effect-triggered ligand changing on the surface of the Ag$_2$S CDs along with pH changing, resulting in luminescence change of Ag$_2$S QDs unit in the hybrid Ag$_2$S CDs, which is inconsistent with the previous report that the luminescence of QDs is ligand dependent [30–32]. Table 1 lists the comparison of this method with some previously reported method for pH sensing in ratiometric method.

Then the reversibility of the Ag$_2$S CDs probes was investigated by changing the pH of Ag$_2$S CDs repeatedly from pH 5.0 to 7.0. Figure 5c illustrated the good reversibility and robustness of Ag$_2$S CDs probes. Next, the selectivity of Ag$_2$S CDs probes towards pH was tested by performing control experiments using CaCl$_2$, MnCl$_2$, BSA, FeCl$_3$, NaCl, KCl and glucose as interferences. No distinct change of the fluorescence ratio (I$_{500}$/I$_{670}$) has been observed, as shown in figure 5d, which suggests the selectivity of the ratiometric probes to pH.

# 3. Conclusion

In conclusion, a novel ratiometric fluorescent probe has been developed for reversible detection of pH with advantage of selectivity. The probe is based on the hybrid structure of green-emission and red-emission Ag$_2$S CDs nanohybrids, and based on the fact that the red-emission fluorescence of the Ag$_2$S CDs can be quenched by increasing pH, whereas the green-emission fluorescence of the Ag$_2$S CDs remains constant. These Ag$_2$S CD-based pH probes are relatively simple to synthesize without requiring complicated steps and expensive instruments, but take advantage of the remarkable photophysical property of Ag$_2$S QDs and CDs, such as excellent photostability, low cost, great safety and remarkable reversibility. These alternative merits endow this simple sensing system a further potential for reliable sensing of pH in food, soil, water and biological samples.

# 4. Experimental section

## 4.1. Materials

Silver nitrate, nitric acid (65%), isopropanol, glutathione (GSH), NaH$_2$PO$_4$·2H$_2$O, ethylenediamine, sodium chloride, ferric chloride hexahydrate, glucose, manganese chloride tetrahydrate, calcium chloride and potassium chloride were obtained from Sinopharm Chemical Reagent Co., Ltd (Shanghai, China). Bovine serum albumin (BSA) and citric acid (99.5%) were acquired from Shanghai Bo'ao Biological Technology Co., Ltd (Shanghai, China). Na$_2$HPO$_4$·12H$_2$O was purchased from Shanghai Xinhua Chemical Reagent Co., Ltd (Shanghai, China). Sulfur powder was obtained from Tianjin Sea Crystal Fine Chemical Co., Ltd (Tianjin, China). Hydrazine hydrate (N$_2$H$_4$·H$_2$O, 85%) was purchased from Tianjin Bodi Chemical Co., Ltd (Tianjin, China). Most of the chemical reagents used in this work are of analytical grade, while glucose and bovine serum albumin are biological reagents. Ultrapure water (Milli-Q, Millipore, 18.2 MΩ resistivity) was used as the experimental water.

## 4.2. Instruments

High-resolution transmission electron microscope (HR-TEM) measurements were examined on a JEM-2100F microscope (JEOL, Japan) operated at 200 kV. The zeta potential and size were taken using a dynamic light scattering (DLS) Malvern Zetasizer instrument (Malvern, England) and a JEM-2100F microscope. The Fourier transform infrared spectroscopy (FTIR) were carried out on a Nicolet Avatar-330 Fourier transform infrared spectrometer (Thermo Nicolet, USA) in transmission mode with KBr window. The surface groups of Ag$_2$S CDs were further verified on a VG Multilab 2000 X-ray photoelectron spectrometer (XPS, Thermo VG, UK) operated at 120 W. UV-vis spectra were collected using a UV–2450 spectrophotometer (Shimadzu, Japan). The fluorescence spectra were recorded on a FLSP920 spectrometer (Edinburgh Instruments Ltd, UK). The pH values of the solutions were acquired by a PHS-3C pH meter (Shanghai, China).

## 4.3. Synthesis of CDs

CDs were synthesized using previously reported methods [28]. Briefly, 2.1010 g of citric acid and 670.0 µl of ethylenediamine were dissolved in 20.0 ml of ultrapure water to form homogeneous and clear solution under stirring. The solution was then transferred into a 30.0 ml of teon-lined autoclave and maintained at 200°C for 5 h, and then centrifuged at 12 000 r.p.m. for 10 min to remove the large dots, and dialysis for 24 h. The as-prepared CDs were stored at 4°C for further usage.

## 4.4. Synthesis of Ag$_2$S quantum dots–carbon dots

Ag$_2$S QDs and Ag$_2$S CDs were prepared according to the previous reports [36,37]. First, 0.1600 g of sulfur powder and 10.0 ml of hydrazine hydrate were completely dissolved by stirring for 24 h to obtain S-N$_2$H$_4$ solution as S$^{2-}$ source. Second, 100.0 ml of ultrapure water, 0.7680 g of glutathione and 0.1700 g of silver nitrate were added into a three-necked flask under nitrogen atmosphere to dissolve completely and a yellow emulsion was obtained. Then 4.0 ml of 20-fold diluted S$^{2-}$ source was added, and the solution turned orange-red. Isopropanol was added for purification after 30 min. The solution was centrifuged at 8000 r.p.m. for 10 min. This step was repeated three times. Finally, the precipitates were re-dissolved in 50.0 ml of ultrapure water to obtain the orange-red transparent Ag$_2$S QDs and stored at 4°C for further use. In the second step, 100.0 μl of CDs (25.5 mg ml$^{-1}$) were also added into the three-necked flask and the other condition kept the same. The final products were Ag$_2$S CDs.

## 4.5. Optimization of carbon dots concentration

Ag$_2$S CDs were synthesized under the same condition by adding 100.0 μl of carbon dots at the concentration of 17.0, 25.5, 34.0, 42.5 and 51.0 mg ml$^{-1}$. The fluorescence spectra of the Ag$_2$S CDs excited at 450 nm were recorded.

## 4.6. Ag$_2$S carbon dots for pH detection

Briefly, 300.0 μl of Ag$_2$S CDs were mixed with 100.0 μl of phosphate-buffer solutions at pH 5.0 to 7.0, and the volume was adjusted to 1.0 ml with ultrapure water. Subsequently, the fluorescence spectra of the solution excited at 450 nm were recorded on FLSP920 spectrometer. The effects of different pH on the UV-vis spectra of Ag$_2$S CDs were evaluated by mixing 200.0 μl of Ag$_2$S CDs solution and 2.8 ml of water. After adjusting the solution pH with sodium hydroxide solution (1.0 mol l$^{-1}$) or hydrochloric acid, UV-vis spectra at different pH values were measured.

## 4.7. Reversibility analysis

Briefly, 3.0 ml of Ag$_2$S CDs and 1.0 ml of phosphate-buffer solution (0.2 mol l$^{-1}$) at pH 5.0 were mixed in a 15.0 ml of EP tube, followed by recording the emission spectrum from 460 to 800 nm under the excitation of 450 nm. Next, the solution was adjusted to pH 7.0 with sodium hydroxide solution (1.0 mol l$^{-1}$), and the fluorescence spectrum was obtained under the same condition. Then the solution was adjusted back to pH 5.0 with hydrochloric acid solution (1.0 mol l$^{-1}$), and the fluorescence spectrum was performed again. This process was repeated five times.

## 4.8. Selectivity experiment

Eight EP tubes were serially numbered, and Ag$_2$S CDs (0.9 mg ml$^{-1}$) and PBS (pH 7.0) were mixed in each tube, followed by the addition of BSA (0.1 mmol l$^{-1}$), glucose (10.0 mmol l$^{-1}$), NaCl (1.0 mmol l$^{-1}$), KCl (1.0 mmol l$^{-1}$), CaCl$_2$ (0.1 mmol l$^{-1}$), FeCl$_3$ (0.1 mmol l$^{-1}$) and MnCl$_2$ (0.1 mmol l$^{-1}$), respectively, to tubes from 2 to 8. The first tube was used as a blank control. The fluorescence spectra of the solutions were recorded under excitation at 450 nm. The ratios of fluorescence intensity of the solution at 500 and 670 nm were plotted on the vertical axis.

Data accessibility. All the necessary data are included in the main manuscript and figures, and raw data for all the figures in the paper have been uploaded as the electronic supplementary material.

Authors' contributions. X.L. and Y.F. designed this study, carried out the laboratory experiments and participated in the data analysis. X.L. wrote the manuscript. Y.W., L.C. and J.L. contributed to the decision of the experimental plan. Y.W., L.C., J.L. and X.L. revised the manuscript. All authors gave final approval for publication.

Competing interests. We declare we have no competing interests.

Funding. The authors express their sincere gratitude for substantial funding about this study received from the Fundamental Research Funds for the Central Universities (grant no. 2662018QD042), National Natural Science Foundation of China (grant no. 31772785) and Da Bei Nong Group Promoted Project for Young Scholar of HZAU (grant no. 2017DBN009).

Acknowledgements. The paper was made better thanks to useful comments and suggestions from three anonymous reviewers and Dr Laura Smith.

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
