## [Reviewer comments · Royal Society Open Science]

Review History

RSOS-200482.R0 (Original submission)

Review form: Reviewer 1

Is the manuscript scientifically sound in its present form?

Yes

Are the interpretations and conclusions justified by the results?

Yes

Is the language acceptable?

Yes

Do you have any ethical concerns with this paper?

No

Have you any concerns about statistical analyses in this paper?

No

Recommendation?

Accept with minor revision (please list in comments)

Comments to the Author(s)

In this manuscript, a novel pH sensor has been developed based on the Ag₂S quantum dots - carbon dots nanohybrids.

The results are very interesting. In my opinion, I recommend the manuscript for publication in RSOS after some modifications.

- 1、 In the Fig. 4, the concentration of Ag₂S quantum dots - carbon dots nanohybrids should be provided.
- 2、 In the Fig. 5, the concentrations of Ag₂S quantum dots - carbon dots nanohybrids and interferences should be provided.
- 3、 In the introduction, some references about ratiometric fluorescent sensors based carbon dots should be cited and discussed.
- 4、 Some discussion about advantages and disadvantages should be added in the discussion part.
- 5、 The following references is advised to add in the manuscript.
 - (1) Nemati, R.; Zare-Dorabei, R., A ratiometric probe based on Ag₂S quantum dots and graphitic carbon nitride nanosheets for the fluorescent detection of Cerium. *Talanta* 2019, 200, 249-255.
 - (2) Cai, M. F.; Ding, C. P.; Wang, F. F.; Ye, M. Q.; Zhang, C. L.; Xian, Y. Z., A ratiometric fluorescent assay for the detection and bioimaging of alkaline phosphatase based on near infrared Ag₂S quantum dots and calcein. *Biosensors & bioelectronics* 2019, 137, 148-153.
 - (3) Cai, M. F.; Ding, C. P.; Cao, X. Y.; Wang, F. F.; Zhang, C. L.; Xian, Y. Z., Label-free fluorometric assay for cytochrome c in apoptotic cells based on near infrared Ag₂S quantum dots. *Anal. Chim. Acta* 2019, 1056, 153-160.
 - (4) Cui, J.; Kim, G.; Kim, S.; Kwon, J. E.; Park, S. Y., Ultra-pH-Sensitive Small Molecule Probe Showing a Ratiometric Fluorescence Color Change. *ChemPhotoChem*. 2020, DOI: 10.1002/cptc.202000023
 - (5) Liu, Z.; Li, G. P.; Wang, Y. N.; Li, J. L.; Mi, Y.; Zou, D. P.; Li, T. S.; Wu, Y. J., Quinoline-based ratiometric fluorescent probe for detection of physiological pH changes in aqueous solution and living cells. *Talanta* 2019, 192, 6-13.

Review form: Reviewer 2

Is the manuscript scientifically sound in its present form?

Yes

Are the interpretations and conclusions justified by the results?

Yes

Is the language acceptable?

Yes

Do you have any ethical concerns with this paper?

No

Have you any concerns about statistical analyses in this paper?

No

Recommendation?

Major revision is needed (please make suggestions in comments)

Comments to the Author(s)

The manuscript presents a ratiometric fluorescent nanoprobe for pH sensing based on Ag₂S-CDs. Some problems should be resolved before publication in the journal.

1. The reason for choosing 25.5 mg/mL should be explained in detailed.
2. Practical applications should be presented.

3. The performance should be compared with other reported results.
4. Could the probe be used to monitor the cellular pH change?
5. How about the stability of the probes?
6. As we all know that there are some functional groups on the surface of CDs, why did the fluorescence of CDs not change with pH?

Review form: Reviewer 3

Is the manuscript scientifically sound in its present form?

Yes

Are the interpretations and conclusions justified by the results?

Yes

Is the language acceptable?

Yes

Do you have any ethical concerns with this paper?

No

Have you any concerns about statistical analyses in this paper?

No

Recommendation?

Major revision is needed (please make suggestions in comments)

Comments to the Author(s)

This manuscript reported a ratiometric fluorescent probe for reversible pH detection based on the hybrid structure of green- and red-emission Ag₂S-CDs nanohybrids. The pH selectivity of Ag₂S-CDs probe also was tested by controlling experiments. I recommend it to be published on Royal Society Open Science after the following major revisions. Some suggestions and comments on this work are listed for the authors:

1. What is the particle size of CDs? What about the morphology of CDs?
2. How can the author prove that the combination of CDs and Ag₂S QDs is the same as the author indicated in Scheme 1?
3. Whether the CDs prepared in this paper exhibit characteristic excitation wavelength-dependent emission.
4. It is better to indicate which sample is analyzed in the caption of XPS spectrum.
5. Page 4 Line 7, author said "25.5 mg/mL was chosen as the optimum CDs concentration for the synthesis of Ag₂S-CDs", but it's "25.5 μg mL⁻¹" in Fig. 5. What is the concentration of CDs? Why the author choose this concentration?
6. Page 4 Line 20-22, please explain this sentence "Similar type of results did not appear for single CDs if Ag₂S QDs and CDs were physically mixed".
7. Fig.5 B, with the change of pH value, the fluorescence peak of red emission shifts. Please explain.
8. Is pH detection limited to 5.0-7.0 for Ag₂S-CDs nanohybrids? Can it be used for a wider range of pH detection?
9. Please pay attention to the spelling of the words in the manuscript, such as "reversible" is "reversible" in Conclusions.

Decision letter (RSOS-200482.R0)

Dear Dr Wu:

Title: A ratiometric fluorescent probe for pH detection based on Ag₂S quantum dots - carbon dots nanohybrids

Manuscript ID: RSOS-200482

The editor assigned to your manuscript has now received comments from reviewers. We would like you to revise your paper in accordance with the referee and Subject Editor suggestions which can be found below (not including confidential reports to the Editor). Please note this decision does not guarantee eventual acceptance.

Please submit your revised paper before 16-May-2020. Please note that the revision deadline will expire at 00.00am on this date. If we do not hear from you within this time then it will be assumed that the paper has been withdrawn. In exceptional circumstances, extensions may be possible if agreed with the Editorial Office in advance. We do not allow multiple rounds of revision so we urge you to make every effort to fully address all of the comments at this stage. If deemed necessary by the Editors, your manuscript will be sent back to one or more of the original reviewers for assessment. If the original reviewers are not available we may invite new reviewers.

RSC Associate Editor:
Comments to the Author:
(There are no comments.)

RSC Subject Editor:
Comments to the Author:
(There are no comments.)

Reviewers' Comments to Author:
Reviewer: 1

Comments to the Author(s)

In this manuscript, a novel pH sensor has been developed based on the Ag₂S quantum dots - carbon dots nanohybrids.

The results are very interesting. In my opinion, I recommend the manuscript for publication in RSOS after some modifications.

- 1、 In the Fig. 4, the concentration of Ag₂S quantum dots - carbon dots nanohybrids should be provided.
- 2、 In the Fig. 5, the concentrations of Ag₂S quantum dots - carbon dots nanohybrids and interferences should be provided.
- 3、 In the introduction, some references about ratiometric fluorescent sensors based carbon dots should be cited and discussed.
- 4、 Some discussion about advantages and disadvantages should be added in the discussion part.
- 5、 The following references is advised to add in the manuscript.
(1) Nemati, R.; Zare-Dorabei, R., A ratiometric probe based on Ag₂S quantum dots and graphitic carbon nitride nanosheets for the fluorescent detection of Cerium. *Talanta* 2019, 200, 249-255.
(2) Cai, M. F.; Ding, C. P.; Wang, F. F.; Ye, M. Q.; Zhang, C. L.; Xian, Y. Z., A ratiometric fluorescent assay for the detection and bioimaging of alkaline phosphatase based on near infrared Ag₂S quantum dots and calcein. *Biosensors & bioelectronics* 2019, 137, 148-153.
(3) Cai, M. F.; Ding, C. P.; Cao, X. Y.; Wang, F. F.; Zhang, C. L.; Xian, Y. Z., Label-free fluorometric assay for cytochrome c in apoptotic cells based on near infrared Ag₂S quantum dots. *Anal. Chim. Acta* 2019, 1056, 153-160.
(4) Cui, J.; Kim, G.; Kim, S.; Kwon, J. E.; Park, S. Y., Ultra-pH-Sensitive Small Molecule Probe Showing a Ratiometric Fluorescence Color Change. *ChemPhotoChem*. 2020, DOI: 10.1002/cptc.202000023
(5) Liu, Z.; Li, G. P.; Wang, Y. N.; Li, J. L.; Mi, Y.; Zou, D. P.; Li, T. S.; Wu, Y. J., Quinoline-based ratiometric fluorescent probe for detection of physiological pH changes in aqueous solution and living cells. *Talanta* 2019, 192, 6-13.

Reviewer: 2

Comments to the Author(s)

The manuscript presents a ratiometric fluorescent nanoprobe for pH sensing based on Ag₂S-CDs. Some problems should be resolved before publication in the journal.

1. The reason for choosing 25.5 mg/mL should be explained in detailed.
2. Practical applications should be presented.

3. The performance should be compared with other reported results.
4. Could the probe be used to monitor the cellular pH change?
5. How about the stability of the probes?
6. As we all know that there are some functional groups on the surface of CDs, why did the fluorescence of CDs not change with pH?

Reviewer: 3

Comments to the Author(s)

This manuscript reported a ratiometric fluorescent probe for reversible pH detection based on the hybrid structure of green- and red-emission Ag₂S-CDs nanohybrids. The pH selectivity of Ag₂S-CDs probe also was tested by controlling experiments. I recommend it to be published on Royal Society Open Science after the following major revisions. Some suggestions and comments on this work are listed for the authors:

1. What is the particle size of CDs ? What about the morphology of CDs?
2. How can the author prove that the combination of CDs and Ag₂S QDs is the same as the author indicated in Scheme 1 ?
3. Whether the CDs prepared in this paper exhibit characteristic excitation wavelength-dependent emission.
4. It is better to indicate which sample is analyzed in the caption of XPS spectrum.
5. Page 4 Line 7, author said " 25.5 mg/mL was chosen as the optimum CDs concentration for the synthesis of Ag₂S-CDs", but it's "25.5 μ g ml⁻¹" in Fig. 5. What is the concentration of CDs? Why the author choose this concentration ?
6. Page 4 Line 20-22, please explain this sentence "Similar type of results did not appear for single CDs if Ag₂S QDs and CDs were physically mixed".
7. Fig.5 B, with the change of pH value, the fluorescence peak of red emission shifts. Please explain.
8. Is pH detection limited to 5.0-7.0 for Ag₂S-CDs nanohybrids? Can it be used for a wider range of pH detection?
9. Please pay attention to the spelling of the words in the manuscript, such as "reversable" is "reversible" in Conclusions.

Author's Response to Decision Letter for (RSOS-200482.R0)

See Appendix A.

Decision letter (RSOS-200482.R1)

Dear Dr Wu:

Title: A ratiometric fluorescent probe for pH detection based on Ag₂S quantum dots - carbon dots nanohybrids

Manuscript ID: RSOS-200482.R1

It is a pleasure to accept your manuscript in its current form for publication in Royal Society Open Science. The chemistry content of Royal Society Open Science is published in collaboration with the Royal Society of Chemistry.

RSC Associate Editor
Comments to the Author:
(There are no comments.)

Reviewer(s)' Comments to Author:

Appendix A

Dear Editors,

Thank you very much for your E-mail dated on April 23, 2020 forwarding the reviewers' comments on our manuscripts titled "A ratiometric fluorescent probe for pH detection based on Ag₂S quantum dots - carbon dots nanohybrids" (**Manuscript: RSOS-200482**). Those comments are all valuable and very helpful for improving our manuscript. According to your suggestions and the reviewer's comments, we have made revision to the manuscript point by point.

Note: We have added more discussion in the revised supporting information. A few more references were added in the revised manuscript. All the added sections were highlighted in the revised manuscript.

The revisions and responses to the reviewers' comments are itemized as follows:

Response to Reviewers:

Reviewer: 1

Comment: In this manuscript, a novel pH sensor has been developed based on the Ag₂S quantum dots - carbon dots nanohybrids. The results are very interesting. In my opinion, I recommend the manuscript for publication in RSOS after some modifications.

Response: We are sincerely appreciated for your positive comment towards to our work. Your comments are valuable and very helpful for improving our manuscript. We hope you best at this unusual time. The revisions according to your comments are as follows:

1. In the Fig. 4, the concentration of Ag₂S quantum dots - carbon dots nanohybrids should be provided.

Response: Thank you for your kind suggestion. The concentration of Ag₂S-CDs in the Fig.4 is 0.02 mg/mL. Please see the legend for Fig. 4 in the revised manuscript.

2. In the Fig. 5, the concentrations of Ag₂S quantum dots - carbon dots nanohybrids and interferences should be provided.

Response: Thank you for your kind suggestion. The concentration of Ag₂S-CDs in the Fig.4 is 0.02 mg/mL. Please see the legend for Fig. 5 in the revised manuscript. The interferences have been provided. We have added it in the revised manuscript.

3. In the introduction, some references about ratiometric fluorescent sensors based on carbon dots should be cited and discussed.

Response: Thanks for your professional suggestion. We have added references [19-21] and discussion part in the revised manuscript. Please see Page 1, right column, the first paragraph in the revised manuscript.

4. Some discussion about advantages and disadvantages should be added in the discussion part.

Response: Thanks for your suggestion. We have added the advantage and disadvantage in the discussion part in the revised manuscript (Page 1, right column, the second paragraph): “The disadvantage of this probe is narrow pH response. However, it shows fluorescence change within a 0.2 pH unit change between 5.0-7.0 with satisfied linear relationship. It is known even pH changes below 1.0 unit, our environments and physiology will be significantly impacted. Hence, our probes, which sharply response to small pH changes within a narrow range are desired.”

5. The following references is advised to add in the manuscript.

(1)Nemati, R.; Zare-Dorabei, R., A ratiometric probe based on Ag₂S quantum dots and graphitic carbon nitride nanosheets for the fluorescent detection of Cerium. *Talanta* 2019, 200, 249-255.

(2)Cai, M. F.; Ding, C. P.; Wang, F. F.; Ye, M. Q.; Zhang, C. L.; Xian, Y. Z., A ratiometric fluorescent assay for the detection and bioimaging of alkaline phosphatase based on near infrared Ag₂S quantum dots and calcein. *Biosensors & bioelectronics* 2019, 137, 148-153.

(3)Cai, M. F.; Ding, C. P.; Cao, X. Y.; Wang, F. F.; Zhang, C. L.; Xian, Y. Z., Label-free fluorometric assay for cytochrome c in apoptotic cells based on near infrared Ag₂S quantum dots. *Anal. Chim. Acta* 2019, 1056, 153-160.

(4)Cui, J.; Kim, G.; Kim, S.; Kwon, J. E.; Park, S. Y., Ultra-pH-Sensitive Small Molecule Probe Showing a Ratiometric Fluorescence Color Change. *ChemPhotoChem*. 2020,DOI: 10.1002/cptc.202000023

(5)Liu, Z.; Li, G. P.; Wang, Y. N.; Li, J. L.; Mi, Y.; Zou, D. P.; Li, T. S.; Wu, Y. J., Quinoline-based ratiometric fluorescent probe for detection of physiological pH changes in aqueous solution and living cells. *Talanta* 2019, 192, 6-13.

Response: Thanks for your kind suggestion. We have added the advised references in the revised manuscript.

Reviewer: 2

Comments: The manuscript presents a ratiometric fluorescent nanoprobe for pH sensing based on Ag₂S-CDs. Some problems should be resolved before publication in the journal.

Response: We are sincerely appreciated for your positive comments towards to our work. Your comments are valuable and very helpful for improving our manuscript. We hope you best at this unusual time. The revisions according to your comments are as follows:

1. The reason for choosing 25.5 mg/mL should be explained in detailed.

Response: Thanks for your comment. It's should be 25.5 μ g/mL, we have corrected in the revised manuscript. The emission peak at 500 nm from CDs was used as the reference signal, the emission peak at 670 nm was used as the signal report unit for pH. Hence, the fluorescence intensity at 500 nm should not be stronger than that at 670 nm. Actually, from Fig. 5A, 17.0 μ g/mL of CDs is also OK, but we chose 25.5 μ g/mL. We have added the reason in the revised manuscript (please see page 3, left column, the second paragraph).

2. Practical applications should be presented.

Response: Thanks for your professional comment. We totally agree with you that practical applications should be presented. We have done early attempts. We tried to monitor the cellular pH change, but no satisfied data have gotten, then we had our winter vacation. I am so sorry that during this special time, our lab has not opened yet to move this practical application forward. Sorry again!

3. The performance should be compared with other reported results.

Response: Thanks for your professional comment. We have compared with other reported results in the revised manuscript. Please see Table 1 in the revised manuscript.

Table 1 Comparison of this method with some previously reported method for pH sensing in ratiometric method

Method	λ_{ex}/nm	λ_{em}/nm	^a Range	^b Ref.
Small molecule probe	405	588/531	6.35-8.0	9
Fluorescent nanogel loaded with fluorophores	450	500/620	4.9-9.0	33
Water-soluble cationic porphyrin derivatives	417	650/719	2.1-8.0	34
Mesoporous silica nanoparticles with rhodamine dye	488	514/550	3.5-7.5	35
Ag ₂ S-CDs nanohybrids	450	500/610	5.0-7.0	This work

a: Linearity range of pH detection

b: Reference

4. Could the probe be used to monitor the cellular pH change?

Response: Thanks for your professional comment. We tried to monitor the cellular pH change, but no satisfied data have gotten. We could only get the conclusion that our probes can be taken into cells by endocytosis, then we had our winter vacation. I am so sorry that during this special time, our lab has not opened yet to move this experiment forward. Sorry again!

5. How about the stability of the probes?

Response: The probes could be stored at 4 °C for at least one month.

6. As we all know that there are some functional groups on the surface of CDs, why did the fluorescence of CDs not change with pH?

Response: Thank you for your comment. We used citric acid and ethylenediamine as precursors to synthesize CDs according to ref.[33] (Zhu et al., *Angew. Chem. Int. Edit.*, 2013, 52, 3953-3957). In this literature, it reported that the synthesized CDs is pH-dependent photoluminescent (PL) behavior. PL intensities decrease in a solution at high or low pH, but remain constant in a solution of pH 4-11. Our detection range is pH 5-7. The reason why the fluorescence of CDs not change with pH maybe is that pH 4-11 is not enough to change the surface functional groups.

Reviewer: 3

Comments: This manuscript reported a ratiometric fluorescent probe for reversible pH detection based on the hybrid structure of green- and red-emission Ag₂S-CDs nanohybrids. The pH selectivity of Ag₂S-CDs probe also was tested by controlling experiments. I recommend it to be published on Royal Society Open Science after the following major revisions. Some suggestions and comments on this work are listed for the authors:

Response: We are sincerely appreciated for your professional comments towards to our work. Your comments are valuable and very helpful for improving our manuscript. We hope you best at this unusual time. The revisions according to your comments are as follows:

1. What is the particle size of CDs? What about the morphology of CDs?

Response: We are so sorry that we have not yet gotten a good TEM image of CDs because of the interference of carbon film when taking TEM image. According to the ref. [33]: Zhu et al., *Angew. Chem. Int. Edit.*, 2013, 52, 3953-3957, the particle diameter of CDs is 2-6 nm. The sizes of CDs were also measured by atomic force microscopy (AFM), and the average height

is 2.81 nm. Most of the morphology of CDs is sphere.

2. How can the author prove that the combination of CDs and Ag₂S QDs is the same as the author indicated in Scheme 1?

Response: Thanks for your so rigorous comment. We have not found out the combination of CDs and Ag₂S QDs. We guess it is core-shell structure, but we have not enough demonstration. It is under research because it is a very interesting phenomenon, and we hope to figure it out. We have re-drawn the Scheme 1, please see it in the revised manuscript.

3. Whether the CDs prepared in this paper exhibit characteristic excitation wavelength-dependent emission.

Response: Yes. We used citric acid and ethylenediamine as precursors to synthesize CDs according to ref.[33] (Zhu et al., *Angew. Chem. Int. Edit.*, 2013, 52, 3953-3957), the CDs exhibit excitation-dependent emission behavior, which is common in fluorescent carbon materials.

4. It is better to indicate which sample is analyzed in the caption of XPS spectrum.

Response: Thanks for your suggestion. We have added the sample name in the legend of Fig.2 in our revised manuscript.

5. Page 4 Line 7, author said “ 25.5 mg/mL was chosen as the optimum CDs concentration for the synthesis of Ag₂S-CDs” , but it's “25.5 μ g ml⁻¹” in Fig. 5. What is the concentration of CDs? Why the author choose this concentration?

Response: Thank you for your so detailed review work. We are so sorry that we made a mistake. The concentration of CDs is 25.5 μ g ml⁻¹. We have corrected in our revised manuscript. The emission peak at 500 nm from CDs was used as the reference signal, the emission peak at 670 nm was used as the signal report unit for pH. Hence, the fluorescence intensity at 500 nm should not be stronger than that at 670 nm. Actually, 17.0 μ g/mL of CDs is also OK from Fig. 5A, but we chose 25.5 μ g/mL. We have added the reason in the revised manuscript (please see page 3, left column, the second paragraph).

6. Page 4 Line 20-22, please explain this sentence “Similar type of results did not appear for single CDs if Ag₂S QDs and CDs were physically mixed”.

Response: I am so sorry that we made a mistake. This sentence should be: “Similar type of results did not appear if Ag₂S QDs and CDs were physically mixed.” This sentence means

Ag₂S QDs and CDs physically mixing cannot result in the result that the fluorescence intensity at 670 nm was gradually weakened with increasing of pH, while the fluorescence intensity at 500 nm remained constant as the internal reference. We have corrected this sentence in the revised manuscript as follows: “Ag₂S QDs and CDs physically mixing cannot result in the change of the fluorescence intensity only at 670 nm with increasing of pH, while the fluorescence intensity at 500 nm remained constant.”

7. Fig.5 B, with the change of pH value, the fluorescence peak of red emission shifts. Please explain.

Response: Because the fluorescent property (including intensity and maximum emission wavelength) of Ag₂S QDs can be tuned according to the surface functional group. In this project, the change of pH results in the surface protonation effect-triggered ligand changing on the surface of the Ag₂S-CDs, so the intensity and maximum emission wavelength of Ag₂S-CDs change.

8. Is pH detection limited to 5.0-7.0 for Ag₂S-CDs nanohybrids? Can it be used for a wider range of pH detection?

Response: It is not limited to 5.0-7.0 for Ag₂S-CDs nanohybrids. However, the I₅₀₀/I₆₇₀ ratio showed good linearity with different pH values only in this range.

9. Please pay attention to the spelling of the words in the manuscript, such as “reversable” is “reversible” in Conclusions.

Response: We are sincerely appreciated for your so strict review work. We are so sorry. We have paid attention to all of the words spelling.